# Balancing Risks and Benefits: Sodium-Glucose Cotransporter 2 Inhibitors and the Risk of Diabetic Ketoacidosis

**DOI:** 10.3390/metabo14030162

**Published:** 2024-03-13

**Authors:** Jan P. Kleinjan, Justin Blom, André P. van Beek, Hjalmar R. Bouma, Peter R. van Dijk

**Affiliations:** 1Department of Endocrinology, University Medical Center Groningen, 9700RB Groningen, The Netherlands; 2Department of Acute Internal Medicine, University Medical Center Groningen, 9700RB Groningen, The Netherlands

**Keywords:** sodium-glucose cotransporter 2 inhibitors, diabetic ketoacidosis

## Abstract

Sodium-glucose cotransporter-2 inhibitors (SGLT2is) are a new class of drugs that have been proven beneficial in the management of diabetes, chronic kidney disease, and heart failure and in the mitigation of cardiovascular risk. The benefits of SGLT2i therapy have led to the rapid adoption of these drugs in clinical guidelines. Since the introduction of these drugs, concerns have arisen, as diabetic ketoacidosis (DKA) unexpectedly occurred in patients treated with SGLT2i. DKA is an infrequent but serious complication of SGLT2i therapy, and is potentially preventable. The risk factors for the development of SGLT2i-associated DKA are inappropriate dose reductions of insulin, the dietary restriction of carbohydrates, and factors that may increase insulin demand such as excessive alcohol intake and major surgery. Moreover, the risk of SGLT2i-associated DKA is higher in persons with type 1 diabetes. It is crucial that both patients and healthcare providers are aware of the risks of SGLT2i-associated DKA. In an effort to encourage safe prescribing of this effective class of drugs, we present two cases that illustrate the risks of SGLT2i therapy with regard to the development of DKA.

## 1. Introduction

Sodium-glucose cotransporter 2 inhibitors (SGLT2is) are effective drugs in the management of type 2 diabetes as they improve glycaemic control, promote weight loss, and mitigate cardiovascular risk and the progression of chronic kidney disease [1]. SGLT2is have also been proven beneficial in the treatment of heart failure, regardless of patients’ glycaemic status [2]. Because of the beneficial cardiovascular and renal effects, this class of oral drugs is even used in type 1 diabetes for selected patients [3]. SGLT2is promote the renal excretion of glucose by inhibiting sodium and glucose reabsorption in the proximal tubule of the kidney, leading to lower plasma glucose levels.

After the introduction of SGLT2is for the treatment of type 2 diabetes, concerns arose as diabetic ketoacidosis (DKA) unexpectedly occurred in persons treated with SGLT2is. DKA is characterised by a triad of hyperglycaemia, ketosis, and metabolic acidosis. It results from a deficiency of insulin as well as an excess of counter-regulatory hormones including glucagon, cortisol, catecholamines, and growth hormone. The state of insulin deficiency results in impaired glucose utilisation by peripheral tissues, thereby stimulating fatty-acid oxidation, resulting in ketone production and acidosis. Hyperglycaemia develops as counter-regulatory hormones stimulate gluconeogenesis and glycogenolysis. DKA most often complicates type 1 diabetes (>67% of cases), but it can also occur in persons with type 2 diabetes [4].

DKA is a rare but serious complication of treatment with SGLT2is [5,6]. The use of SGLT2is can predispose individuals to the development of euglycemic DKA (arbitrarily defined as DKA with blood glucose < 14 mmol/L), as their inhibiting effect on renal glucose and sodium reabsorption results in glucosuria, lower plasma glucose levels, decreased insulin production, and, consequently, the induction of lipolysis and ketogenesis. The degree of hyperglycaemia is a result of the balance between hepatic glucose production and glucosuria (see Figure 1). The risk of SGLT2i-associated DKA is particularly high in persons with type 1 diabetes. Therefore, the use of this class of drugs in this patient group is still controversial, especially because large cardiovascular outcome trials on SGLT2i use in type 1 diabetes are lacking. 

The significant benefits of SGLT2i therapy have led to the rapid adoption of these drugs in diabetes guidelines, as well as cardiovascular and renal guidelines. Hence, clinicians will increasingly be confronted with SGLT2i-associated adverse events including (euglycaemic) DKA. It is, therefore, vital that clinicians, both within and outside the field of diabetes, are aware of the risks of SGLT2i therapy and the factors that may elicit DKA. We present two cases that illustrate that these risk factors are insufficiently addressed in clinical practice. Balancing benefits and risks in daily practice is complicated as it differs per indication and type of patient it is used for. Although there is abundant literature on the DKA risk and cardiorenal benefits associated with SGLT2i therapy, a brief overview of the risks and benefits across different patient groups is lacking. We provide a narrative review and concisely summarise data from meta-analyses of randomised controlled trials in patients with and without diabetes (type 1 and type 2). By presenting these data collectively, we hope to facilitate the safe and effective prescribing of this class of drugs.

## 2. Case Presentations

### 2.1. Case 1

A 46-year-old woman was admitted to the emergency department of the University Medical Center Groningen (The Netherlands) in 2022 after she had been found in her home in a comatose state. Her medical history included obesity (BMI 38 kg/m^2^), hypertension, and type 2 diabetes since the year before presentation. She was treated with metformin, gliclazide, semaglutide, and dapagliflozin. Upon examination, she was unresponsive. A nasopharyngeal airway was inserted to maintain an open airway. Oxygen saturation was 95% with 2 L/min of oxygen supplementation through a nasal canula. Her blood pressure was 110/70 mmHg with a heart rate of 150 beats per minute. Neurological examination revealed a reduced Glasgow Coma Scale (E1M3V1). Her body temperature was 34.1 °C and a distinct acetone smell was noticed. Blood analysis revealed severe high anion gap metabolic acidosis, pronounced hyperkalaemia, and hyperglycaemia, as well as acute renal insufficiency and marked ketonuria (see Table 1).

Based on the findings upon clinical examination and the high anion gap acidosis with hyperglycaemia and ketonuria, a diagnosis of severe diabetic ketoacidosis (DKA) with hyperkalaemia (due to a combination of an intracellular shift and reduced excretion secondary to acute kidney injury) was made. Treatment was initiated with calcium gluconate as a membrane stabiliser, fluid resuscitation (normal saline and bicarbonate 8.4% to lower potassium), and a continuous infusion of insulin (2 IE/h).

She was admitted to the intensive care unit (ICU) for further treatment of the DKA with intravenous insulin and fluid resuscitation. All oral glucose-lowering medications were withheld. At that moment, the exact cause of the DKA was still unclear. The patient recovered quickly and was discharged to the general ward the next day. Given the relatively low incidence of DKA among patients with type 2 diabetes, serological analysis was performed to assess the presence of antibodies indicative of type 1 diabetes, which came back negative. Additional history revealed that the patient had recently been encouraged to lose weight by adhering to a diet low in carbohydrates. Two weeks before presentation, her general practitioner co-prescribed the SGLT2i dapagliflozin because the treatment goal of an HbA1c < 53 mmol/mol was not reached. Treatment with insulin was stopped as it did not seem to have a sufficient effect. The combination of these risk factors led to the development of DKA. Upon hospital discharge, insulin therapy was continued, and treatment with semaglutide and metformin was reinitiated. Dapagliflozine was not restarted.

### 2.2. Case 2

A 48-year-old male patient was admitted to the emergency department of the University Medical Center Groningen (The Netherlands) in 2023 because of complaints of nausea and vomiting. He had a medical history of poorly controlled type 1 diabetes mellitus (most recent HbA1c 64 mmol/mol) with micro- and macrovascular complications including ischemic cardiomyopathy and peripheral artery disease. Six months prior to presentation, the SGLT2i dapagliflozine was initiated by the cardiologist to optimise the management of heart failure. He had been ill for three days with complaints of abdominal pain, loss of appetite, and frequent vomiting. He reported that he had continued to use his subcutaneous insulin, as his blood glucose levels had been higher than usual. Upon admission, the patient was drowsy but responsive, tachypnoeic with normal oxygen saturation, and there were clinical signs of volume depletion. Further physical examination revealed a necrotic ulcer on his right hallux and heel. Blood analysis revealed metabolic acidosis with an increased anion gap of 32 mmol/L and normal lactate levels. Further analysis to determine the cause of this high anion gap revealed an extremely high acetone concentration (see Table 1). The venous glucose concentration was only mildly elevated. Concentrations of ethanol and toxic alcohols (ethanol and ethylene glycol) were undetectable in serum. Based on these findings, we concluded that this patient had euglycaemic DKA, potentially triggered by a reduced intake of food and the use of an SGLT2i in this patient with type 1 diabetes.

Rehydration therapy with normal saline and intravenous insulin was initiated. A glucose 5% solution was administered to enable continuous insulin infusion. Within 12 h of treatment, the patient’s bicarbonate levels normalised. Over the course of the next two days, his mental status improved and complaints of nausea resolved. Treatment with dapagliflozin was withdrawn permanently.

## 3. Discussion 

We present two case reports that illustrate the risks of DKA in patients treated with an SGLT2i. In clinical trials, the incidence rates of DKA varied from 0.2 to 2.2 events per 1000 patient years in persons with type 2 diabetes, a 2.5-fold increase compared to placebo [5]. This might be an underestimation of real-world data where incidence rates up to 4.9 events per 1000 patient years are reported [5,7]. In type 1 diabetes, the risk of SGLT2i-associated DKA is substantially higher (see Table 2). Based on randomised controlled trials, the risk for DKA may be six to eight times higher for persons with type 1 diabetes treated with an SGLT2i compared to placebo. Real-world incidence rates in type 1 diabetes are estimated to be as high as 43–71 events per 1000 patient years [8]. 

The clinical manifestations of SGLT2i-associated DKA are similar to DKA occurring due to other causes, although the blood glucose levels may be lower than anticipated as a result of SGLT2i-induced glucosuria (see Figure 1). Although the absolute risk of DKA in patients treated with SGLT2i is low, a combination of factors may predispose individuals to developing this metabolic complication. These risk factors include inappropriate dose reductions or the omission of insulin, the dietary restriction of carbohydrates, and factors that may increase insulin demand such as excessive alcohol intake and major surgery. All of these risk factors cause a relative deficiency of insulin, leading to lipolysis and ketogenesis [6]. These factors are potentially preventable and should be considered by every healthcare provider who prescribes SGLT2is, as illustrated by the two cases presented here.

The patient described in case 1 developed life-threatening DKA requiring ICU admission after the initiation of an SGLT2i in combination with the abrupt cessation of insulin therapy, while on a low-carbohydrate diet and using a GLP-1 receptor analogue (GLP-1RA). The combination of these risk factors presumably contributed to the development of DKA: the cessation of subcutaneous insulin causes a disbalance in the total insulin-to-glucagon ratio, stimulating lipolysis and, thereby, ketogenesis. This is further aggravated by the low supply of dietary carbohydrates and SGLT2i-induced glucosuria. Finally, the simultaneous use of a GLP-1RA might have contributed to the development of the DKA [11]. The underlying relationship between GLP-1RA use and DKA remains unclear, but might be the result of gastro-intestinal side effects of GLP-1RA including nausea, vomiting, and diarrhoea, resulting in dehydration, as well as decreased carbohydrate intake due to loss of appetite. The development of this severe DKA may have been prevented if her insulin was tapered slowly and normal carbohydrate intake was advised before starting dapagliflozin. Although a diet low in carbohydrates does not increase the risk of DKA itself, the combination of SGLT2i treatment with a low carbohydrate diet is associated with an increased risk of DKA.

Prescribing SGLT2i in a patient using insulin may require reducing the insulin dose in order to prevent hypoglycaemia, as this is a known adverse effect of SGLT2i. However, there is scarce literature to support an empirical reduction in insulin dose after the start of an SGLT2i. A meta-analysis from 2017 showed an average reduction in insulin requirement of 8.8 units after initiating an SGLT2i [12]. When starting an SGLT2i, a maximum insulin dose reduction of 10–20% is appropriate, followed by frequent dose readjustments based on glucose monitoring. When a significant decrease in insulin dose is necessary, caution is warranted and self-monitoring of ketones during acute illness might be helpful to screen for DKA [13,14].

In case 2, we describe a man with type 1 diabetes who developed euglycaemic DKA after he was prescribed dapagliflozin for the management of heart failure. The increased risk for DKA has led to the rejection of market authorisation for the use of SGLT2is in persons with type 1 diabetes by the FDA. The European Medicines Agency, however, has approved low-dose dapagliflozin (5 mg) and sotagliflozin (200 mg) for patients with a body mass index of 27 kg/m^2^ or higher [3], mainly because of the reported benefits on glycaemic control. The potential cardiorenal benefits of SGLT2i were not considered at the time of market authorisation for the management of type 1 diabetes, as data on cardiorenal outcomes were lacking. It is important to note that the glycaemic benefits of SGLT2is are modest, especially compared to the major cardiorenal benefits that are reported in patients with and without type 2 diabetes. Studies suggest that these protective cardiorenal effects might also apply to persons with type 1 diabetes. A retrospective cohort study involving persons with type 1 diabetes treated with insulin in combination with SGLT2is or GLP-1RA for at least 6 months reported cardiorenal benefits. The SGLT2i-treated cohort showed the preservation of the estimated glomerular filtration rate (eGFR) over a 5-year period compared with the GLP-1 RA-treated cohort (+3.5 vs. −7.2 mL/min/1.73 m^2^), including patients with established chronic kidney disease (CKD). Moreover, patients treated with SGLT2is were less likely to develop heart failure (relative risk of 0.44) and CKD (relative risk 0.49) when compared to GLP-1RA-treated patients [15]. A post hoc analysis of the DEPICT trials, two randomised controlled trials investigating the use of dapagliflozin in type 1 diabetes, showed a significant reduction in albuminuria after 1 year of dapagliflozin treatment compared to placebo. The studied subgroup of patients with baseline albuminuria (>30 mg/g) had a reduction of 13 and 31% of albuminuria for dapagliflozin 5mg and 10mg, respectively. However, no difference in eGFR was noted between the subgroups [16].

The current evidence does not conclusively support the routine prescription of SGLT2 inhibitors in type 1 diabetes. Weighing the risks and benefits of SGLT2 inhibition in patients with type 1 diabetes remains a challenge for clinicians as there are major safety concerns, but sufficient clinical data on cardiovascular and renal benefits are lacking. Table 2 summarises the risks for DKA as well as the major cardiorenal benefits, as reported in large randomised placebo-controlled trials for patients with and without diabetes. The lack of cardiovascular outcome data in type 1 diabetes makes clinicians hesitant to prescribe SGLT2i therapy for this patient group. However, the possible harm of withholding a class of drugs with such an impressive potential for the prevention of cardiovascular and renal damage should be seriously considered. This highlights the need for renal and cardiovascular outcome trials of SGLT2i therapy in patients with type 1 diabetes. Until such data are available, the prescription of SGLT2is in type 1 diabetes may be considered for patients at high risk for cardiovascular or chronic kidney disease, if the risk for DKA is minimised by careful patient selection, education, and monitoring. Consultation with an endocrinologist or diabetologist is warranted before initiating SGLT2i therapy in patients with type 1 diabetes. Practical algorithms are available that can aid in the safe prescription of SGLT2is as well as dose adjustments of glucose-lowering co-medication [17]. Furthermore, all patients using SGLT2is should be informed about the risks of DKA and carefully instructed about precipitating factors and management strategies. For instance, the STICH protocol (Stop SGLT2i, Inject Insulin, consume 30grams or less of Carbohydrates, Hydrate), is a helpful and easy way to instruct both patients and healthcare professionals about the management of SGLT2i-associated DKA [13].

## 4. Conclusions

The use of SGLT2is is increasing both in hospital and within the primary care setting, as indications for these drugs are expanding because of the established major cardiovascular and renal benefits. SGLT2i-related DKA is rare, but can be life-threatening and may be preventable. The risks and benefits of SGLT2i therapy should be carefully balanced. We aim to support safe and effective prescribing and create awareness among all healthcare professionals about the risk of DKA associated with SGLT2i therapy, especially since its potential euglycaemic presentation may hamper the timely recognition of the DKA. Insulin therapy should never be stopped acutely when prescribing SGLT2is, and caution is warranted when reducing the insulin dose in patients at risk for developing DKA. Weight loss strategies should not include strict carbohydrate restriction. Patients with type 1 diabetes are at an even higher risk of developing DKA when using SGLT2is. Healthcare providers and patients should be instructed about what precautionary measures can be taken to prevent DKA during treatment with SGLT2is. In this way, the maximal therapeutic potential of this effective class of drugs can be optimally utilised.

## Figures and Tables

**Figure 1 metabolites-14-00162-f001:**
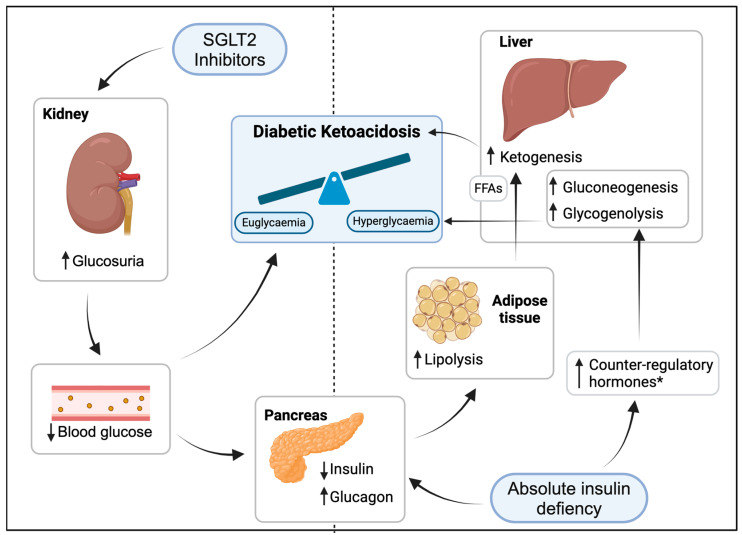
Pathophysiology of SGLT2i-associated diabetic ketoacidosis. * cortisol, catecholamines, growth hormone. FFAs: free fatty acids.

**Table 1 metabolites-14-00162-t001:** Laboratory results upon admission.

	Case 1	Case 2	Reference Value
pH	6.67	7.28	7.35–7.45
Bicarbonate (mmol/L)	4	14	24–28
Sodium (mmol/L)	138	142	135–145
Chloride (mmol/L)	107	96	97–107
Potassium (mmol/L)	8.2	4.3	3.5–5.0
Creatinine (umol/L)	160	73	45–80
Albumin (g/L)	41	43	35–50
Anion gap	27	32	8–12
Lactic acid (mmol/L)	1.3	2.2	<2.2
Glucose (mmol/L)	35	13.7	<7.8
Acetone (mg/L)	n.d.	739	1–20 *
Urinary ketones	3+	3+	Negative

* in fasting state up to 50 mg/L, n.d.: not determined.

**Table 2 metabolites-14-00162-t002:** Overview of DKA risk and major benefits of SGLT-2 inhibitors in patients with and without diabetes, based on randomised placebo-controlled trials.

	Type 1 Diabetes [9]	Type 2 Diabetes [10]	No Diabetes [2]
**Major risk**			
Diabetic ketoacidosis	5.8	2.2	n.a.
Absolute DKA risk (events/1000 p.y.)	40	0.8	0
**Major benefits**			
All-cause mortality	i.d.	0.85	0.88 *
Cardiovascular death	i.d.	0.84	0.85
Hospital admissions for heart failure	i.d.	0.69	0.72
Composite renal endpoint **	i.d.	0.55	0.64

Risks and benefits as compared to placebo are reported as hazard ratios unless otherwise specified. Patients without diabetes are patients with heart failure or chronic kidney disease. p.y. = patient years. i.d. = insufficient data. n.a. not applicable. * Not significantly lower than placebo. ** Progression of kidney disease, renal replacement therapy, or death from renal cause.

## Data Availability

No new data were created or analyzed in this study. Data sharing is not applicable to this article.

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
