# Peer review of "Balancing Risks and Benefits: Sodium-Glucose Cotransporter 2 Inhibitors and the Risk of Diabetic Ketoacidosis"

_metabolites, 2024, doi:10.3390/metabo14030162_

Round 1

Reviewer 1 Report

Comments and Suggestions for Authors

Dear authors,

I have reviewed your work, and I find it very interesting. However, I have some observations that, from my perspective, could help improve its presentation.

In the introduction, the molecular mechanism of action triggered by SGLT2 inhibitors should be integrated.

In lines 59 and 94, the third person should be used, indicating the name and location of the hospital, as well as the timeframe in which the patients were recruited.

In Table 1, the authors should specify whether values of acetone were not determined, were undetectable, or if the values were lost.

In line 183, the authors use the expression "In summary," however, it is not the appropriate place to provide a summary.

The presented results are rather straightforward; the authors should compare the data more extensively, allowing for a partial clarification of the various mechanisms involved.

The authors should include perspectives and clinical impact that their work could denote.

Author Response

Please see attachment (Response to Reviewer 1 Comments)

Reviewer 2 Report

Comments and Suggestions for Authors

This two-case study is about consequences of application of sodium-glucose cotransporter 2 inhibitors. 

First of all, I would like to refer to the title. Although, I understand that the title should be catching to increase citing however, from my perspective it is not necessary to use the first free words. 

The paper is written very clearly, the description of the cases gives enough information for the reader, I have no remarks to discussion. The main problem in this paper is its aim. I am not convinced that these two cases are so unique that they are worth publishing, especially that authors refer to many studies confirming the described phenomena. So, from my perspective there's no point in publishing these two-case study until authors will not underline the uniqueness of these cases and underline a clear aim of this study besides “creating awareness among all health care professionals”. 

Besides:

Either remove "FFA’s: free fatty acids" from Figure 1 caption or add it to the figure

Add letters n.d. (and explain under the table “not determined”) in Case 1, line Acetone

Numbers and units should be written separately (e.g. line 180-181).

Round 2

Reviewer 2 Report

Comments and Suggestions for Authors

You convinced me that this work is worth publishing but what you wrote in response 2 should be put in the introduction, especially the part concerning lack of data which are collectively presented.
